# Giant Paratesticular Liposarcoma: Molecular Characterization and Management Principles with a Review of the Literature

**DOI:** 10.3390/diagnostics12092160

**Published:** 2022-09-06

**Authors:** Giuliana Pavone, Chiara Romano, Federica Martorana, Lucia Motta, Lucia Salvatorelli, Antonio Maria Zanghì, Gaetano Magro, Paolo Vigneri

**Affiliations:** 1Division of Medical Oncology, A.O.U. Policlinico “G. Rodolico–San Marco”—Catania, Via Santa Sofia, 78, 95123 Catania, Italy; 2Center of Experimental Oncology and Hematology, A.O.U. Policlinico “G. Rodolico–San Marco”—Catania, Via Santa Sofia, 78, 95123 Catania, Italy; 3Department of Medical and Surgical Sciences and Advanced Technology G. F. Ingrassia, A.O.U. Policlinico “G. Rodolico–San Marco”—Catania, Via Santa Sofia, 87, 95123 Catania, Italy; 4Department of Clinical and Experimental Medicine, University of Catania, 95123 Catania, Italy

**Keywords:** paratesticular liposarcoma, well-differentiated liposarcoma, rare tumors, *MDM2*, *CDK4*

## Abstract

Paratesticular liposarcomas are extremely rare malignant tumors originating from fat tissues, with an often-challenging diagnosis. We present here the case of a 76-year-old man with a giant paratesticular liposarcoma, initially misdiagnosed as a scrotal hernia. After two years, the progressively enlarging mass underwent surgical resection, and a diagnosis of well-differentiated liposarcoma (lipoma-like subtype) was made. Post-operative treatments were not indicated, and the patient remains relapse free. Next generation sequencing performed on the neoplastic tissue showed co-amplification of *MDM2* and *CDK4*. These alterations are molecular hallmarks of well-differentiated liposarcomas and corroborate the histological diagnosis. Clinical and molecular features of the presented case are in line with the majority of previously published experiences. In conclusion, the presence of a liposarcoma should be taken into account during the diagnostic workup of scrotal masses, in order to minimize the rate of misdiagnosis and improper management. Molecular analysis may support histological characterization of these rare entities and potentially disclose novel therapeutic targets.

Liposarcomas (LS) represent 20% of all soft tissue sarcomas (STS) [1]. There are three main LS subgroups: well-differentiated/dedifferentiated LS (WDL/DDL), myxoid/round cell LS (MRCL) and pleomorphic LS (PLS) [2]. Well-differentiated liposarcomas and DDL are the most common histological variants, accounting for over 60% of all LS cases. Typically, WDL display a low replication index, with slow growth and low metastatic rates, while DDL represents the progression of WDL from an indolent, sometimes locally aggressive lesion to a more rapidly growing disease with metastatic potential [3]. Well-differentiated LS and DDL are almost universally associated with alterations involving chromosome 12q, which carries the oncogenes *Murine Double Minute 2* (*MDM2*), *Cyclin Dependent Kinase 4* (*CDK4*) and *High-Mobility Group AT-Hook 2* (*HMGA2*) [3]. In particular, *MDM2* and *CDK4* are frequently co-amplified in *WDL/DDL*, as well as more than 75% of LS present *HMGA2* amplification [4]. Otherwise, *HMG2A* rearrangements are sometimes encountered in mesenchymal tumors, including lipomas [5], typically after the first three exons encoding the AT-hook domains, which determine an in-frame fusion transcript or gene truncation [6].

Next generation sequencing (NGS) identifies these genetic hallmarks often contributing to the correct diagnosis [7,8].

The retroperitoneum and limbs represent the most frequent LS presentation sites [9], while scrotal localizations are very uncommon [10]. Indeed, paratesticular LS are rare pathological entities, with approximately 200 cases described thus far and only few reports of giant paratesticular LS (i.e., measuring over 10 cm) [11,12,13,14]. Usually, paratesticular LS presents as a painless inguinal or scrotal swelling, indistinguishable from benign masses such as hernias, hydroceles, orchitis, scrotal lipomas or epidermoid cysts. Therefore, the diagnosis is often challenging until surgical resection. Due to their rarity, recommendations on paratesticular LS management are based on evidence derived from previously reported cases. Currently, orchifuniculectomy with wide local resection is the standard of care for localized disease. No consensus exists about adjuvant treatments (i.e., chemotherapy or radiation therapy) despite the high relapse rates [14,15,16,17].

Here we present the case of a giant paratesticular LS for which we performed NGS analysis. We also provide a brief literature review on the clinical, pathological and molecular features of this entity and its current therapeutic indications, as well as future perspectives.

In October 2017, a healthy 76-year-old man presented with a painless right scrotal mass that had been slowly growing over a period of two years. On physical examination, the lesion displayed parenchymatous-soft consistency and could be distinguished from the right testicle. Serum beta-human chorionic gonadotropin (b-HCG), lactate dehydrogenase (LDH) and alpha-fetoprotein (AFP) were within the normal range. Ultrasonography (US) showed an extra-testicular mass of about 5 cm which was considered a scrotal hernia. Therefore, hernioplasty was proposed to the patient, but he refused surgery. In December 2019, as his inguinal discomfort worsened, the patient underwent another US, which revealed a significant growth of the scrotal mass, now measuring 10 cm, and excluded the presence of a hernial orifice. According to these findings, he was diagnosed with a giant primary scrotal lipoma and underwent surgical resection of the mass. Due to the size of the tumor and its adherence to surrounding tissues, an en bloc excision was not feasible and the lesion was removed in nine fragments (Figure 1).

Macroscopically, the tumor appeared as a solid mass of adipose tissue with heterogeneous consistency and a yellowish lipoma-like cut surface. Histopathologic examination revealed a lesion composed of mature fat with variably sized adipocytes separated into lobules by bland fibrous septa. Cellularity was low and mitotic figures were uncommon. Atypical spindle cells in fibrous/fibromyxoid stroma, or adjacent to vessels were seen. Rare lipoblasts were found. No heterologous differentiation was identified. Immunohistochemically, neoplastic cells showed positivity for MDM2 and CDK4 (Figure 2).

Based on morphological and immunohistochemical features, a diagnosis of well differentiated liposarcoma, lipoma-like subtype was made. Given this diagnosis, a contrast-enhanced computed tomography (CT) of the thorax, abdomen and pelvis was performed to exclude the presence of distant metastases. The patient then underwent rescue orchiectomy with high ligation of the spermatic cord and a wide excision. Adjuvant radiation therapy or chemotherapy were considered unnecessary. After a 30-month follow-up he is in good condition with no evidence of disease recurrence. Considering the rarity of the case, we decided to investigate the molecular profile of the tumor. Hence, we extracted both DNA and RNA from the formalin-fixed paraffin-embedded (FFPE) tumor tissue and sequenced them employing two NGS panels: (1) a DNA custom panel identifying point mutations, deletions/insertions or copy number variations (CNV) in 34 genes associated with STSs; and (2) the FusionPlex Expanded Sarcoma Panel (ArcherDx), which uses RNA as input material to look for key fusions and variants in 63 genes relevant for sarcomagenesis. Libraries were sequenced on the Ion GeneStudioTM S5 Plus sequencer and analyzed with the Ion ReporterTM software, version 5.18 (Thermo Fisher Scientific) and with the Archer Analysis software, version 6.2 (ArcherDx).

Sequencing analysis identified molecular alterations specific for WDL, thus confirming the histological diagnosis. Indeed, we found the co-amplification of *MDM2* and *CDK4* on chromosome 12q14-15, with a copy number variation of 10.1×. We also retrieved two gene fusions involving *HMGA2*: (1) the *HMGA2-UNC5D* fusion, involving exon 3 on *HMGA2* and exon 5 on *UNC5D*, was retrieved in 52% of the sequences; (2) the *HMGA2-LOC102724030* fusion, involving exon 3 on *HMGA2* and exon 2 on *LOC102724030*, was found in the 18% of the sequences. As expected, we found no *FUS-DDIT3* rearrangements, *PIK3CA* mutations or deletions of 13q which would have suggested an MRCL or a PLS, respectively [18,19].

We report here the case of a man diagnosed with a giant paratesticular LS, with typical WDL molecular alterations.

Scrotal LS are frequently misdiagnosed. Physical examination and US are unable to discriminate these entities from lipomas, especially in the case of small or well-differentiated tumors with homogeneous fatty patterns and slow growth rates that can be misinterpreted as benign features [11,20,21,22,23]. Pre-operative CT and/or MRI can provide relevant information and should always be considered in doubtful cases [24].

Furthermore, it may be challenging to histologically distinguish a well-differentiated liposarcoma, lipoma-like, from lipoma. Typically, lipoma is superficial and lacks atypical nuclei. Hence, in deep-seated lesions (retroperitoneum, pelvis or abdomen), recurrent neoplasms, older patients with deep extremity lesions (>10 cm) and in cases with bland nuclei or ambiguous interpretation of atypia, hybridization techniques (amplifications of *MDM2* and *CDK4*) are strongly suggested for diagnosis confirmation [25,26].

Overall, delayed or sub-optimal treatments are frequent due to these characteristics [12,27]. The clinical history of our patient is in line with the majority of reported cases, which presented a slowly enlarging scrotal mass treated as a benign lesion and diagnosed as a LS after histological examination (Table 1).

Considering the risk of mistakes, a diagnosis of paratesticular LS must be taken into account during the diagnostic workup of scrotal masses, regardless of their clinical presentation, echogenicity pattern and growth rate. Therefore, upfront wide local excision with radical orchiectomy represents the preferred option in these cases. According to the general consensus, wide resection with ipsilateral orchiectomy and spermatic cord excision is the standard of care, while locoregional or retroperitoneal lymphadenectomy should be reserved for patients with evidence of lymphatic disease [27]. Radical surgery with clear margins is the most relevant factor to reduce the risk of local recurrence. Indeed, positive margins are associated with a 3-year recurrence-free survival of 29% compared to 100% in the case of negative margins [29]. The role of adjuvant chemotherapy or radiation therapy remains controversial [11,12]. They can be considered in case of an estimated high recurrence risk, as in subjects with positive excision margins or high histological grade [12]. However, each case should be discussed by a multidisciplinary sarcoma board before making a final decision that will then be discussed with the patient [27]. Regardless of the possible risk factors and the treatment received, paratesticular liposarcomas always require long-term follow-up, due to the remarkable rate of local recurrence and distant metastases [11,12,14,15,23,28].

In the case of relapsed/advanced disease, chemotherapy and radiotherapy exert a different role according to the histological variant (WDL/DDL, MRCL and PLS) [19,30]. Doxorubicin monotherapy, or in combination with ifosfamide are the regimens of choice in the first-line setting. Since WDL/DDL are usually resistant to standard systemic therapies, surgical re-resection, when feasible, is a viable approach for recurrent disease. On the other hand, MRCL and PLS are more chemo- and radio-sensitive. Hence, surgical resection of oligo-metastases, palliative radiation and systemic treatments are all potential alternatives [3]. In this complex framework, molecular characterization of LS may be useful for both the diagnostic and therapeutic workup. Indeed, the identification of specific molecular hallmarks can be useful for the differential diagnosis between LS and lipoma [26,31,32]. In particular, the co-amplification of *MDM2* and *CDK4* is a hallmark of WDL/DDL, while *HMGA2* rearrangements are sometimes encountered in benign lesions (such as lipomas) and in other types of mesenchymal tumors [4,5,33].

Additionally, *CDK4* amplification may represent a therapeutic target in WDL/DDL. To date, several trials evaluated the efficacy of CDK4/6Is as single agents or in combination with other agents [34]. Associations of CDK4/6Is with MDM2 inhibitors (HDM201) or mTOR inhibitors (everolimus) seem particularly promising [35]. Several studies suggest the usefulness of transcriptomic analysis in the differential diagnosis between WDLfrom lipomas and DDL from other sarcomas by comparing these data with those publicly available from The Cancer Genome Atlas [36,37]. Overall, the diagnosis and treatment of WDL/DDL poses several challenges, especially in the case of atypical presentations, such the paratesticular ones, of which clinicians should be aware. Innovative technologies for molecular analysis should be fully exploited in order to provide the best possible management for patients with this rare disease.

## Figures and Tables

**Figure 1 diagnostics-12-02160-f001:**
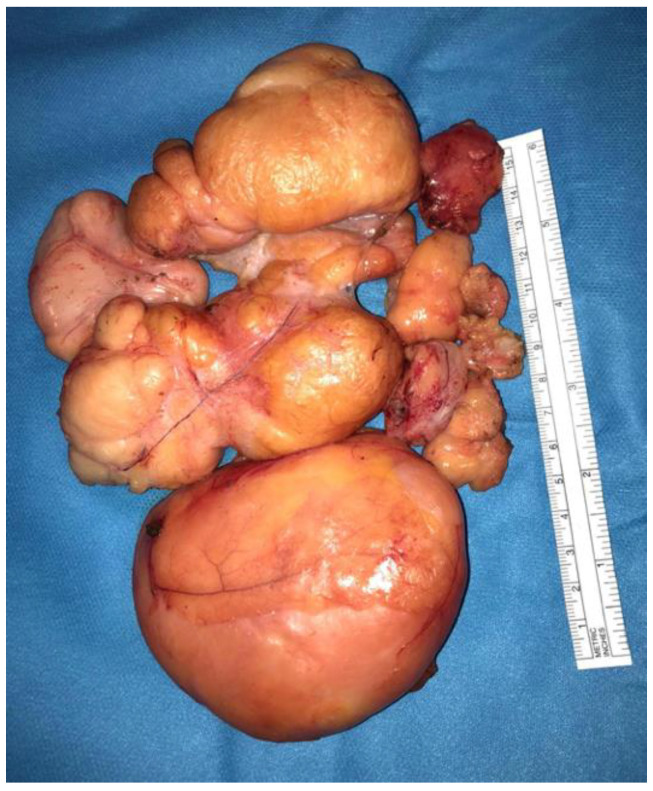
Macroscopic findings on the surgical specimen. A solid mass of yellowish adipose tissue was removed in fragments, the major with a maximum diameter of 14.5 cm.

**Figure 2 diagnostics-12-02160-f002:**
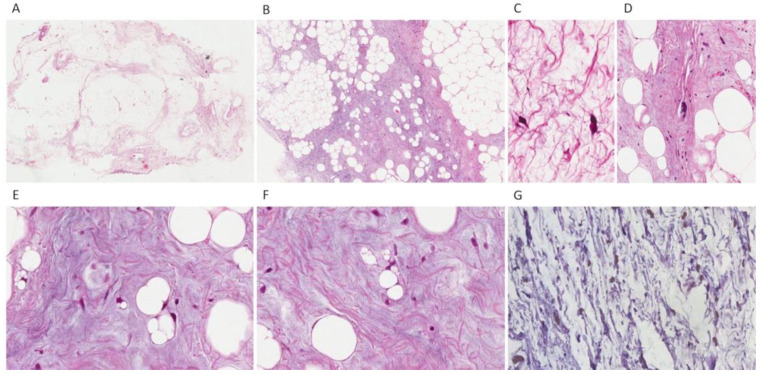
Pathological findings. Histopathologic examination: (**A**) a lobulated lesion composed of mature fat; (**B**) variably sized adipocytes set in fibrous/fibromyxoid stroma; (**C**,**D**) atypical cells with enlarged, hyperchromatic nuclei in fibro-myxoid septa; (**E**,**F**) multivacuolated lipoblasts with atypical nuclei. (**G**) Immunohistochemical analysis showed nuclear positivity for MDM2 in neoplastic cells.

**Table 1 diagnostics-12-02160-t001:** Case reports of primitive paratesticular liposarcomas.

Author	Type	Age	Growth Time	Size (cm)	Side	Pain	Initial Diagnosis	Primary Surgery	Margins	CT Scan	Post-Surgical Treatment(s)
Zuwei (2018) [11]	DDL	51	2 m	8	Right	No	Spermatocytoma	Orchiectomy	R0	NA	None
Sopena-Sutil (2016) [12]	WDL	56	25 y	40 × 40	Left	No	Liposarcoma	Orchiectomy and spermatic cord ligation	R0	Negative	None
Alyousef (2013) [14]	WDL	75	6 y	8.5 × 5.4	Right	No	Suspicious scrotal mass	Orchiectomy and inguinal canal contents resection	R0	Negative	None
Grossi (2014) [15]	WDL/MRCL	81	4 y	28 × 30	Right	No	Suspicious scrotal mass	Orchiectomy and high spermatic cord ligation	NA	Negative	None
Kalyvas (2004) [20]	WDL	72	5 y	10 × 9	Left	No	Inguinal hernia	Orchiectomy and spermatic cord ligation	NA	Negative	None
Omidvari (2014) [21]	WDL	55	30 y	NA	Left	No	Scrotal lipoma	Tumor resection	R1	Negative	Re-surgery and RT
Li (2013) [22]	WDL/MRCL	53	2 y	5.5 × 4.2	Left	No	Inguinal hernia	Orchiectomy, spermatic cord ligation and inguinal lymph node biopsy	R0	Negative	None
Keenan (2019) [23]	DDL	82	1.5 m	11 × 9	Left	Yes	Scrotal hematoma	Hemiscrotectomy	R1	Positive (Pelvis)	Palliative RT
Keenan (2019) [23]	WDL	54	24 y	3 × 3	Left	Yes	Inguinal hernia	Tumor resection	NA	Positive (Lung)	NA
Ayari (2018) [28]	MRCL	67	8 m	4	Right	No	Suspicious scrotal mass	Orchiectomy and high spermatic cord ligation	NA	Negative	None

Legend: WDL: well-differentiated liposarcoma; DDL: dedifferentiated liposarcoma; MRCL: myxoid/round cell liposarcoma; m: months; y: years; R0: no residual disease after primary surgery; R1: residual disease after primary surgery; RT: radiotherapy; NA: not available.

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
