# Peer review of "Giant Paratesticular Liposarcoma: Molecular Characterization and Management Principles with a Review of the Literature"

_diagnostics, 2022, doi:10.3390/diagnostics12092160_

Round 1

Reviewer 1 Report

I would appreciate you giving me an opportunity to review this manuscript. The authors mentioned usefulness of next generation sequencing to detect MDM2/CDK4 amp, HMGA2 rearrangement for liposarcoma. As the authors mentioned, genomic analysis can provide useful information. I thus think that this manuscript can be informative and acceptable after addressing some points.

I suppose there are some points to address.

1.     The authors did not show radiographic images such as enhanced CT or MRI. I am aware of the fact that not only histological finding but also genomic examination is important, however, radiographic information is also essential for appropriate diagnosis. In this setting, I wonder how important HMGA2 rearrangement is for diagnosis. In general, MRI can clearly detect well differentiated liposarcoma. I ask the authors to show radiographic images.   

2.     Which is more reliable one for appropriate diagnosis, MDM2/CDK4 amp or HMGA2 rearragement? Since we regularly check MDM2/CDK4 amp in clinic and recognize usefulness of them, I wonder whether HMGA2 rearrangement can give more useful information for diagnosis or not? Statement is necessary in Discussion Part.  

3.     (Page2, line 53) What does the word ‘abonout’ mean?

4.     Do any malignant or benign tumors have HMGA2 arrangement other than paratesticular liposarcoma? I ask the authors to give information in Discussion Part.

Author Response

  1. As described, the patient was initially misdiagnosed with scrotal lipoma and no pre-operative staging was performed. He underwent post-operative staging with CT and MRI. Hence imaging of the primary tumor is not available. However, we agree with the reviewer about the importance of imaging in the diagnostic workup of WDL. We have added a sentence (lines 128-130 - "Pre-operative CT and/or MRI can provide relevant information and should always be considered in doubtful cases") and a reference (Shim, Eur J Radiol 2020) in the Discussion section to underline this aspect.
  2. We thank the reviewer for this observation. We agree that MDM2/CDK4 co-amplification has a diagnostic and prognostic significance in WDLS, while HMGA2 rearrangements can be retrieved in different types of mesenchymal lesions, including lipomas. Hence, we clarified this concept in the Introduction (lines 41-44: "In particular MDM2 and CDK4 are frequently co-amplified in WDL/DDL, as well as more than 75% of LS present HMGA2 amplification. Otherwise, HMG2A rearrangements are sometimes encountered in mesenchymal tumors, including lipomas") and Discussion section (lines 183-186: Several studies suggest the usefulness of transcriptomic analysis in the differential diagnosis between WDLS from lipomas and DDLS from other sarcomas by comparing these data with those publicly available from The Cancer Genome Atlas), revised the Abstract accordingly and added two new references (Damerell, Signal Transduct Target Ther 2021; Mertens, Genes Chromosomes Cancer 2016).

  3. It is a typing error. We corrected it in “about”.

  4. HMGA2 alteration may occurred in other type of mesenchymal and epithelial neoplasms. We added a sentence about this in the Discussion section (lines 176-179: "In particular, the co-amplification of MDM2 and CDK4 is a hallmark of WDL/DDL, while HMGA2 rearrangements are sometimes encountered in benign lesions (such as lipomas) and in other type of mesenchymal tumors") along with a new reference [Unachukwu, Int J Mol Sci 2020].

Reviewer 2 Report

This report includes a comprehensive clinical and genetic description of an interesting case of paratesticular liposarcoma. I don't have major concern on this study. To utilize the size of the case, it would have been much more interesting if they could sample many areas of the tumor and perform genome/transcriptome analyses. Can the authors include this in the discussion section?

Author Response

We thank the reviewer for this insightful comment. Despite the dimensions of the tumor certainly allow multiple sampling from different areas to perform molecular analyses, we are currently unable to retrieve additional material to carry them out.

Still, we acknowledge the rationale of this observation and decided to comment on it in Discussion section (lines 183-186: "Several studies suggest the usefulness of transcriptomic analysis in the differential diagnosis between WDLS from lipomas and DDLS from other sarcomas by comparing these data with those publicly available from The Cancer Genome Atlas". 

Ref:

- Wang, A Rapid and Cost-Effective Gene Expression Assay for the Diagnosis of Well-Differentiated and Dedifferentiated Liposarcomas. J Mol Diagn 2021, 23, 274-284, doi:10.1016/j.jmoldx.2020.11.011

- Comprehensive and Integrated Genomic Characterization of Adult Soft Tissue Sarcomas. Cell 2017, 171, 950-965.e928, doi:10.1016/j.cell.2017.10.014.

Round 2

Reviewer 1 Report

The authors addressed all issues I pointed out.